# Associations between the School Environment and Physical Activity Pattern during School Time in Swedish Adolescents

**DOI:** 10.3390/ijerph181910239

**Published:** 2021-09-29

**Authors:** Gisela Nyberg, Örjan Ekblom, Karin Kjellenberg, Rui Wang, Håkan Larsson, Britta Thedin Jakobsson, Björg Helgadóttir

**Affiliations:** 1Department of Physical Activity and Health, The Swedish School of Sport and Health Sciences (GIH), 114 86 Stockholm, Sweden; orjan.ekblom@gih.se (Ö.E.); karin.kjellenberg@gih.se (K.K.); rui.wang@gih.se (R.W.); hakan.larsson@gih.se (H.L.); britta.thedinjakobsson@gih.se (B.T.J.); bjorg.helgadottir@gih.se (B.H.); 2Department of Global Public Health, Karolinska Institutet, 171 77 Stockholm, Sweden; 3Division of Clinical Geriatrics, Department of Neurobiology, Care Sciences and Society, Karolinska Institutet, 171 64 Solna, Sweden

**Keywords:** accelerometers, leisure-time, moderate to vigorous physical activity, physical education, school breaks, school policy, school time, students, teachers

## Abstract

Knowledge about associations between school-based initiatives and physical activity patterns is limited. The purpose of this paper was to examine associations between factors in the school environment, physical activity and sedentary time during school time. The cross-sectional study included 1139 adolescents aged 13–14 from 34 schools. Physical activity and sedentary time were measured using hip-worn accelerometers. Factors in the school environment included health policy, a mobile phone ban during breaks, organized physical activities during breaks and activity breaks during lessons reported by teachers. The frequency and duration of breaks and physical education (PE) lessons were collected from school schedules. The results showed significant associations between health policy (β = 3.87, 95% confidence interval (CI): 2.37, 5.23), the mobile phone ban (β = 2.51, 95% CI: 1.29, 3.94) and PE; total duration (β = 0.08, 95% CI: 0.05, 0.11), average duration (β = 0.08, 95% CI: 0.01, 0.13) and frequency (β = 1.73, 95% CI: 0.50, 3.04) and moderate-vigorous physical activity. There were negative associations between health policy (β = −6.41, 95% CI: −10.24, −2.67), the mobile phone ban (β = −3.75, 95% CI: −7.25, −0.77) and PE; total duration (β = −0.15, 95% CI: −0.23, −0.08) and average duration (β = −0.14, 95% CI: −0.27, −0.03) and time spent sedentary. Adolescents attending schools with health policies, mobile phone bans and more time for PE showed higher levels of physical activity and lower sedentary time.

## 1. Introduction

Being physically active during adolescence has substantial benefits for physical and mental health, both in the short and long term [1,2]. Furthermore, children’s physical activity and participation in physical education (PE) classes are of great importance for their physical activity level, fitness and metabolic health even in adulthood [3]. Even so, the majority of adolescents do not meet the international guidelines of 60 min of moderate to vigorous physical activity (MVPA) daily [4], and boys are more physically active compared to girls [5,6,7]. In addition, physical activity levels decrease with increasing age [7,8].

National data from Sweden with adolescents using accelerometry showed that the total sedentary time was on average 10.3 h per day [9]. Results from European children aged 9–12 years showed that the average time spent sedentary was vastly lower, or 8.6 h per day [10]. Also, excessive use of screen time is common all over the world among children and adolescents and is particularly concerning in high and very high-income countries [11]. There is evidence to suggest that among children and adolescents, more time spent in sedentary behavior, in particular recreational screen time, clusters with lower fitness, poorer cardiometabolic health, shorter sleep duration, and unfavorable measures of adiposity [12]. Thus, efforts that promote physical activity and reduce the extensive time spent sedentary among children and adolescents are needed.

In the school setting, most adolescents can be reached irrespective of background characteristics and therefore many schools have implemented physical activities in different forms. Interventions targeting factors in the school environment have yielded promising results, showing that school-based policy including break time length and facilities were associated with physical activity levels in adolescents. However, studies on how sedentary behavior was influenced by school-based policies are scarce [13]. Other school factors that have been shown to be associated to increased physical activity in youth are health promotion policies covering physical activity actions inside and outside the PE classes and improvements in playgrounds, such as repainting and improving physical space [14]. However, no definite conclusions can be drawn due to heterogeneity across studies and lack of prospective data. Of critical importance, a recent meta-analysis concluded that school interventions have not been effective at actually increasing children’s daily minutes of accelerometer-assessed moderate-to-vigorous physical activity, emphasizing the need for better implementation [15]. A better understanding of the factors in the school environment that are associated with physical activity patterns will support the development of effective school-based interventions.

The aim of the study was to examine associations between specific factors in the school environment and physical activity patterns in Swedish adolescents. In addition, the purpose was to study whether the associations differed by sex. The factors in the school environment included health policy, a mobile phone ban during breaks, organized physical activities during breaks, activity breaks during lessons and frequency and duration of breaks and PE lessons.

## 2. Materials and Methods

### 2.1. Study Design, Setting and Participants

The design of the study was cross-sectional. Schools situated within a two to three hour drive around Stockholm, Sweden, were invited by e-mail to participate in the study. Schools with a sports profile, with fewer than 15 students in each class, and those with a student population not speaking Swedish were excluded. In total, 558 schools were invited and 84 schools accepted the invitation. For feasibility reasons, the inclusion of schools stopped after 40 schools had been included based on a variation in type of municipality and socioeconomic background of the schools (see flow chart, Figure 1). In total, 467 schools did not respond and they were not contacted again since the required sample size had been reached. One to four classes participated from each school. All students in the participating classes in grade 7 (aged 13–14 years) were invited to participate. In total, 1556 students were invited and 1139 accepted and provided informed consent (73% participation rate). The participants received a compensation in the form of a gift card. The schools, teachers and parents did not receive any compensation.

### 2.2. Ethical Statement

All the participants and their parents gave their informed consent before participation in the study. The study was conducted in accordance with the Declaration of Helsinki and the protocol was approved by the Ethical Review Agency in Stockholm, Sweden (Dnr: 2019-03579).

### 2.3. Data Collection

The students came by bus or train to the Swedish School of Sport and Health Sciences (GIH) in Stockholm between September and December 2019. The measurements were carried out by trained healthcare professionals and researchers. Physical activity and sedentary time in students were measured by accelerometry. The monitors were distributed by the researchers during their visit to GIH. The students were instructed to wear the accelerometers on the right hip for seven consecutive days when awake, except during water-based activities. After the accelerometers had been worn for seven days, the class teacher collected the monitors and sent them back to researchers at GIH in pre-paid envelopes. Other measures were height, weight and country of birth. In addition, questionnaires were sent to the teachers and the parents. Questions about the school environment were self-reported by teachers and parental education was self-reported by parents.

#### 2.3.1. Physical Activity and Time Spent Sedentary

Physical activity and time spent sedentary were measured with accelerometers (model GT3X+, Actigraph, LCC, Pensacola, FL, USA) for seven consecutive days. The participants were instructed to wear the accelerometer on their right hip during all hours awake, only taking it off for water-based activities i.e., showering or swimming.

The software ActiLife Data Analysis, version 6.13.3 was used to process the accelerometer data. Uniaxial (vertical axis) data were saved in 5 s epochs. Non-wear time was defined as 60 min of 0 counts, with vector magnitude and no spike tolerance. A filter was created to separate school time physical activity from leisure time physical activity by using the school schedules for each class. Schedules were missing for four classes and were replaced with schedules from another class in the same grade from the same school. Data on physical activity patterns for school time were considered valid if there were at least two valid weekdays while the whole week was considered valid if there were at least three valid days where at least one of those days was a weekend day. To determine if a day was valid another filter was used based on sleep time from a self-reported questionnaire. Sleep time was based on self-reported questions about when the participants usually got up and when they usually went to bed for weekdays and weekends, respectively, with response options in 30-min intervals. The days were considered valid if there was at least 500 min of wear time. The first day of registration was not used in the analyses.

The physical activity outcomes were sedentary time, light physical activity (LIPA) and MVPA. The counts from the accelerometer data were categorized into minutes spent in: sedentary intensity (0–100 counts/minute), LIPA (101–2295 counts/minute) and MVPA (≥2296 counts per minute) [16]. The total wear time for school time was calculated based on the start and end of each school day, extracted from the school schedules.

MVPA during leisure time was processed in two steps. Firstly, the MVPA time during weekdays was divided into MVPA during school and MVPA during leisure time using exact times from the school schedules, reported as mean MVPA leisure time per day during weekdays. Secondly, total MVPA during leisure time was calculated by adding MVPA time during the weekend to MVPA during leisure time on weekdays, and then calculated to represent average MVPA leisure time per day during the whole week. Sex-specific tertiles were divided into low, medium and high volume of MVPA during leisure time in order to investigate if there was a difference in associations between physical activity patterns and school factors depending on activity levels outside of school time. The weekly average of ≥60 min of MVPA per day was used to categorize the participants into reaching or not reaching the physical activity recommendation, based on both school and leisure time.

#### 2.3.2. Anthropometry

Body mass (kg) and height (cm) were measured according to standardized procedures using a calibrated scale (Tanita BC-418, Tanita corporation, Tokyo, Japan) to the nearest 0.1 kg and a stadiometer (SECA 5123, SECA Weighing and Measuring Systems) to the nearest mm. Body mass index (BMI) was calculated as body mass (kg) divided by height (m) squared. Underweight, normal weight, overweight and obesity were defined according to the International Obesity Task Force recommendations, with different cut-off values depending on age and sex [17].

#### 2.3.3. School Environment Factors

Teachers from the participating schools were contacted and asked to answer a questionnaire containing questions on their school environment. This was done shortly after their students visited GIH. Teachers from all schools responded to the questionnaire. The number of teachers responding to the questionnaire varied from one to seven teachers per school, with the head teacher and PE teacher being the most frequent respondents. The factors in the school environment included health policy, the mobile phone ban during breaks, organized physical activities during breaks and physical activity breaks during lessons. The questionnaire to the teachers included four questions.

Regarding health policy, the questionnaire included the question “Does your school have a health policy with the aim to work towards increasing physical activity and healthy dietary habits among the students?” The response alternatives “Yes, physical activity” and “Yes, both physical activity and dietary habits” were collapsed into one category and “Yes, dietary habits”, “No neither of those” and “I don’t know” were collapsed into one category.

Regarding mobile phone ban during breaks, the questionnaire included the question “Are students allowed to use their mobile phones during breaks?” The response alternatives were categorized as “Yes” while “No” and “I don’t know” were collapsed. Regarding organized physical activities during breaks, the questionnaire included the question “We have organized physical activities during breaks that are led by a teacher or a student. The responses were categorized as “Every day” while “1–3 times per week” and “Seldom or never” were combined into one category. Regarding physical activity breaks during lessons, the questionnaire included the question “We have activity breaks or other types of physical activities during the lessons (apart from PE lessons)”. The responses were categorized as “Every day” while “1–3 times per week” and “Seldom or never” were combined into one category. For each question, the answers from the teachers from each school were summed up into the following categories: (1) yes (everyone in agreement), (2) disagreement (some chose the most favorable options, while others chose one of the other options), and (3) no (everyone in agreement). For the variable mobile phone ban, there were only two schools where the teachers were not in agreement. In this case, these schools were categorized as “no” as the disagreement group was too small to be analyzed.

Information on frequency and duration of breaks and PE classes were retrieved from the schedules that were collected for each class and included two variables. Regarding breaks, data on the average number of minutes in breaks per day, the average number of minutes per break and the average number of breaks per day were retrieved from the class schedules from each class. Regarding PE, data on the total number of minutes in PE classes per week, average number of minutes per PE class and number of PE classes per week were retrieved from the class schedules from each class

#### 2.3.4. Parental Education

The highest level of education for the mother or father was self-reported in a questionnaire and used as an indicator of socioeconomic status (SES). The variable was dichotomized into low education (≤12 years of schooling) and high education (>12 years of schooling). Additionally, information on the level of parental education in the participating schools were extracted from publicly available data from the Swedish National Agency for Education. On a school level, the proportion of parents with education above >12 years per school was used to describe the schools’ characteristics.

#### 2.3.5. Country of Birth and Foreign Background

The country of birth and foreign background of the students and parents were self-reported in questionnaires. Country of birth was categorized into born in Sweden, born in Europe, and born outside Europe. Foreign background was categorized into born in Sweden with at least one Swedish born parent and born outside of Sweden, or both parents born outside of Sweden.

#### 2.3.6. Municipality

The type of municipality was categorized into one of three main groups based on the Swedish municipality classification 2017. The first two groups, large cities and municipalities near large cities and medium-sized towns and municipalities near medium-sized towns were categorized as urban areas. The last group included smaller towns/urban areas and rural municipalities and was categorized as rural areas [18].

### 2.4. Statistical Analyses

Descriptive statistics are expressed as proportions for categorical variables and means with standard deviations for continuous variables. The different schools can explain a considerable proportion of variance in school time physical activity patterns (30% of MVPA, 43% of sedentary time). Therefore, multi-level linear regression models were used to determine if school environmental factors were associated with physical activity pattern (sedentary time, LIPA and MVPA, respectively), for both school time and leisure time. We have applied the models with two-levels: the school level and the individual/student level. A random intercept was modelled for each school. All analyses were adjusted for sex, parental education and accelerometer wear time. We checked the model fitness and heteroscedasticity by conducting goodness of fit tests, plotting residuals as well as their relationships with predictors. A bootstrapping estimation was performed to produce sensible estimates for standard errors in the multi-level model as suboptimal fitness of model or heteroscedasticity were detected. Additionally, analyses for school time and leisure time were stratified by sex. When significant associations were found between factors in the school environment and physical activity patterns, analyses to check for significant interactions were run. The level of significance was set to *p* < 0.05. The IBM SPSS Statistics for Windows software, version 26 was used for the statistical analyses.

## 3. Results

The total sample included 1139 participants from 34 schools. The sample was equally divided between boys (51.0%) and girls (49.0%) and the mean (SD) age was 13.4 (0.3) years, see Table 1. The proportion of students with at least one parent with more than 12 years of education was 72%. The prevalence of overweight and obesity in the sample was 20%. The schools were mostly from urban areas (88%) and on average 60% of parents of all the students in each school had more than 12 years of education.

Of the whole sample, 903 (79%) of the participants had valid accelerometer registrations for a whole week and 1054 (95%) of the participants had valid registrations for school days. Slightly more girls than boys had valid registrations (*p* < 0.001 for the whole week and *p* = 0.014 for the school week) but no statistically significant differences were found in parental education and BMI. Approximately one-third of the sample met the recommendation of 60 min MVPA per day in this age group (Table 1). The proportion was significantly lower for girls than for boys (25% versus 37%). During school time, girls had less minutes in MVPA compared to boys (*p* < 0.001) but there was no difference between sexes during leisure time (*p* = 0.145), see Figure 2. Girls were more sedentary during school time (*p* < 0.001) and had lower LIPA compared to boys (*p* < 0.001). However, during leisure time, girls had both higher levels of LIPA (*p* < 0.001) and sedentary time (*p* = 0.01) than boys. The variation of MVPA between schools is depicted in Figure 3, where the level of MVPA ranged from 15 to 44 min across schools.

The results showed positive significant associations between health policy (β = 3.87, 95% CI: 2.37, 5.23), mobile phone ban during breaks (β = 2.51, 95% CI: 1.29, 3.94) and PE classes; total duration per week (β = 0.08, 95% CI: 0.05, 0.11), average duration of classes (β = 0.08, 95% CI: 0.01, 0.13) and number of classes per week (β = 1.73, 95% CI: 0.50, 3.04) and MVPA during school time. All the associations except for average duration of physical activity classes were significant for both boys and girls shown in Table 2. Also, total duration of breaks per day was positively associated with MVPA during school time for girls shown in Table 2.

There were significant positive associations between organized physical activity during breaks (β = 4.92, 95% CI: 1.65, 8.72) and total duration of PE classes per week (β = 0.06, 95% CI: 0.004, 0.12) and LIPA during school time. These associations were also significant for boys shown in Table 3. There was a significant negative association shown between activity breaks during lessons and LIPA during school time for girls, as shown in Table 3.

In addition, the results showed significant negative associations between health policy (β = −6.41, 95% CI: −10.24, −2.67), mobile phone ban during breaks (β = −3.75, 95% CI: −7.25, −0.77) and PE classes; total duration per week (β = −0.15, 95% CI: −0.23, −0.08) average duration of classes (β = −0.14, 95% CI: −0.27, −0.03) and time spent sedentary during school time. There were significant negative associations between health policy and PE classes: total duration per week and average duration of classes for boys and between sedentary time during school time and total duration of PE classes per week and sedentary time during school time for girls, as shown in Table 4.

No significant interactions between MVPA tertiles during leisure time and the factors in the school environment were found.

Furthermore, the school factors that were significantly associated with physical activity patterns during school time were also tested in models with physical activity patterns during leisure time as the outcome. There was a significant positive association between mobile phone ban during breaks (β = −12.83, 95% CI: −23.09, −4.06) and MVPA during leisure time. In addition, there was a negative association between mobile phone ban during breaks (β = 4.58, 95% CI: 2.56, 6.49) and sedentary time during leisure time. There were no other significant associations between school factors and physical activity patterns during leisure time.

## 4. Discussion

In this cross-sectional study, the associations between specific factors in the school environment and physical activity and sedentary time in adolescents were studied. Significant associations between physical activity patterns during school time and health policy, a mobile phone ban during breaks, and time in PE were found.

In this study, the results suggest that schools do not have a compensatory effect. The participants were divided into tertiles of MVPA during leisure-time, but the least physically active participants did not seem to benefit more from different physical activity initiatives in school compared to the participants with higher MVPA.

A systematic review has shown that factors in the school environment such as break time length, facilities, and activity settings (type and location) have been associated with adolescents’ physical activity, with limited studies on sedentary behavior [13]. However, the same study also reported that there have been few studies that have tried to modify the school environment for adolescents, and it is therefore difficult to know which types of environmental interventions are effective in secondary schools. Knuth and Hallal have reported that factors in the school environment that have shown to be associated with physical activity levels have been health promotion policies covering physical activity actions inside and outside the PE classes and improvements in playgrounds [14]. The results regarding health promotion policy was supported in a study with Spanish adolescents (*n* = 15,902) aged 11–18 years, where it was found that schools with better policies of promotion of physical activity showed a higher compliance with the physical activity recommendations [19]. In the present study, there were similar results showing associations between the existence of health policies covering physical activity, mobile phone ban and more time for PE with higher levels of physical activity and lower levels of sedentary time. A study by Durant et al. reached a similar conclusion where they showed that PE may be a promising intervention as school days with PE per week was correlated with physical activity in adolescents aged 12–18 years (*n* = 165) from the United States [20]. The findings were also supported in a study with adolescents (*n* = 17,766) from the United States where they showed associations between participation in daily school PE and MVPA [21]. In addition, Erwin et al. concluded in a review that PE was one component that contributed to students’ levels of daily physical activity [22].

The comparability between studies is limited since there is a heterogeneity in methods for evaluating both physical activity patterns and factors in the school environment, including, for example, checklists of spaces, satellite photos, interviews, and questionnaires with teachers and head teachers [14]. Also, the comparability is limited because the purpose and structure of PE differs between countries. Another challenge is that many studies have problems with implementation fidelity of interventions in schools which makes it difficult to evaluate the effects of specific strategies [15,23]. One of the reasons for the lack of implementation is that many teachers experience a high workload where time constraints appear to be a major barrier for the implementation of programmes [24]. Similar results were presented in another review by Naylor et al. where they showed that time was the most prevalent factor influencing implementation [25]. In addition, Hills and authors have presented challenges for physical activity promotion in schools where a crowded school curriculum was one oft-cited barrier [26]. In the present study, the teachers and head teachers reported conflicting results regarding the implementation of different strategies in their schools. One explanation for that could be that the head teachers have decided upon different strategies, but the teachers experience that the strategies are not implemented in their organization. On the other hand, the results from the present study show that there is a large potential for increasing physical activity during school hours, as there was a substantial difference between the participating schools in terms of MVPA.

In the present study, there were similar associations for boys and girls between factors in the school environment and MVPA. Boys showed more significant associations between factors in the school environment and LIPA and sedentary time. Also, there were significant differences between boys and girls regarding the strength of the associations. Health policy and more time in PE might be more impactful for boys while the results of the mobile phone ban suggest that mobile phone bans might be equally important for boys and girls. This is contrary to other studies, where some evidence has been found showing that girls may respond better to physical activity interventions compared to boys [27]. However, several studies have shown that boys were more physically active than girls during breaks in schools [28].

In this study, boys had higher levels of MVPA than girls, and this was only evident during school-time and not during leisure-time. In addition, the girls had more minutes in sedentary time during school-time compared to boys, indicating that the school environment may be more suited towards boys’ physical activity patterns. The results differ from the finding in a Swedish study, Riksmaten Adolescents, which included adolescents aged 11–18 years (*n* = 3302), where there were differences in physical activity patterns between sexes both during school-time and during leisure time [9]. The differences in results may be due to differences in categorizing time in school-time versus leisure-time where the previous study used a set time filter between 08.00 and 15.59 compared to using school schedules for each participant in the present study. This might also explain the fact that in the survey Riksmaten Adolescents, the time in MVPA was higher during school-time compared to leisure-time (31 min/23 min) compared to the present study (26.5 min/32 min). In a meta-analysis including 91 studies (*n* = 42,463, mean age 10.1 years) from 29 countries, the pooled mean of time spent in MVPA was estimated, and showed that approximately 24 min were spent in MVPA during school time and 23.5 min were spent in time after-school [29].

One strength of the study was the large sample size of adolescents. Another strength was the use of accelerometry, which is a valid and reliable method to measure physical activity patterns. Yet another strength was the precise measure of school-time and leisure-time as school schedules were collected for each school class and used for the accelerometry analyzes. Furthermore, a precise measure of total wear time was used (as sleep time was used) for each participant to categorize time into time awake based on their self-reported answers. This present study also had some limitations. One limitation was the self-reported questionnaires, where teachers from the same school responded differently for some questions, which led to conflicting answers. This may have influenced the results. Another limitation was the cross-sectional design of the study. Furthermore, the schools that accepted the invitation to participate may have been more engaged and unique compared to the schools that did not participate. However, both the schools and the participants were quite representative for the population with regard to socioeconomic background and school organization.

## 5. Conclusions

This study showed that adolescents attending schools with health policies, mobile phone bans and more time for PE classes showed higher levels of physical activity and lower sedentary timecompared to schools without policies. There were significant differences between boys and girls in physical activity patterns in favor of the boys regarding health policy and more time in PE while there was no difference for the mobile phone ban amongst the sexes. Well-designed intervention studies are needed to study whether the introduction of such school policies can be effective strategies to increase physical activity and reduce sedentary behavior in adolescents.

## Figures and Tables

**Figure 1 ijerph-18-10239-f001:**
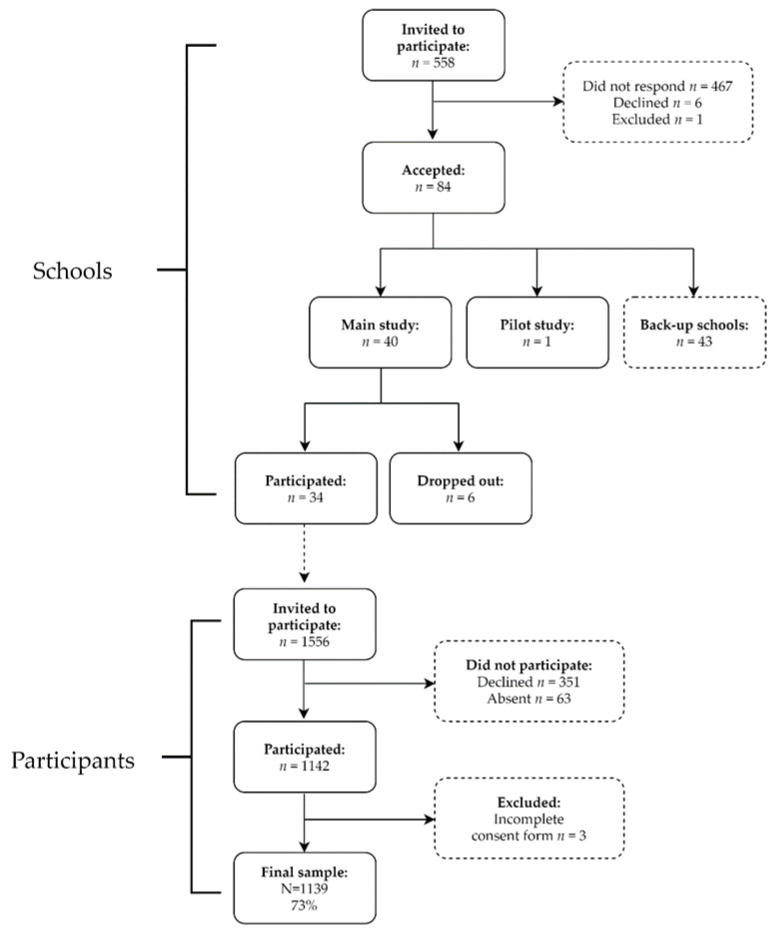
Participant flow diagram.

**Figure 2 ijerph-18-10239-f002:**
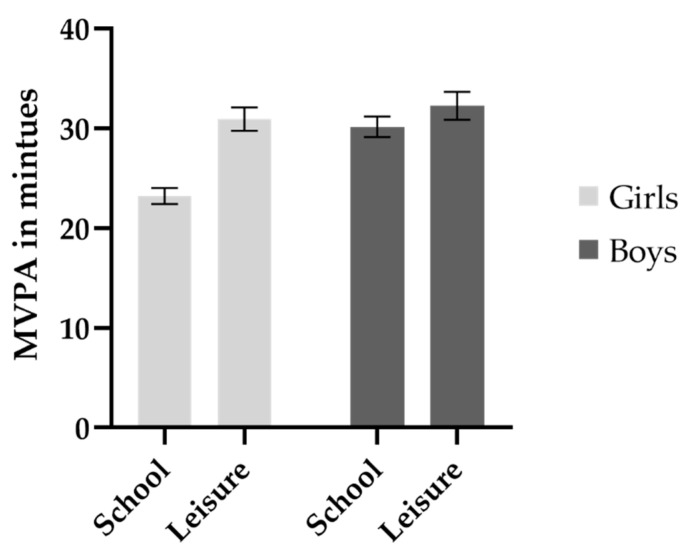
Minutes in moderate-to-vigorous physical activity during school time across sex, presented as means with 95% confidence intervals.

**Figure 3 ijerph-18-10239-f003:**
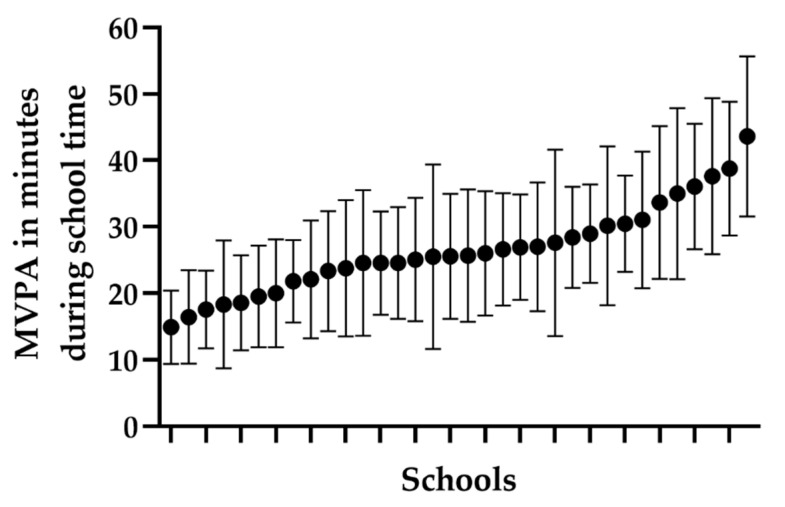
Variation in the participants’ moderate-to-vigorous physical activity (MVPA) during school time across schools in ranked order, presented as means with standard deviation. Schools are ordered after time spent in MVPA.

**Table 1 ijerph-18-10239-t001:** Descriptive statistics for the students and the schools.

Student Level Variables				
	Total	Girls	Boys	Sig.
	*n* (%)	*n* (%)	*n* (%)	*p*
Number of students	1139 (100)	580 (51.0)	558 (49.0)	
Age (mean ± SD)	13.4 ± 0.3	13.4 ± 0.3	13.4 ± 0.4	0.147
Parental education				
≤12 years	275 (28.4)	144 (29)	131 (27.8)	0.674
>12 years	695 (71.6)	353 (71)	341 (72.2)	
Foreign background				
Swedish born, and at least one Swedish born parent	800 (71.6)	414 (72.5)	386 (70.8)	0.233
Born outside Sweden, or both parents born outside of Sweden	317 (28.4)	157 (27.5)	159 (29.2)	
Country of birth				
Sweden	967 (85.7)	490 (84.9)	476 (86.4)	0.758
Europe, including Nordic countries	46 (4.1)	24 (4.2)	22 (4.0)	
Outside Europe	116 (10.3)	63 (10.9)	53 (9.6)	
BMI categories *				
Underweight	89 (7.8)	38 (6.6)	51 (9.2)	0.203
Normal weight	815 (71.8)	430 (74.1)	384 (69.3)	
Overweight	179 (15.8)	89 (15.3)	90 (16.2)	
Obese	52 (4.6)	23 (4.0)	29 (5.2)	
Physical activity during school time (mean ± SD)				
Moderate-to-vigorous physical activity	26.5 ± 11.2	23.2 ± 9.6	30.1 ± 11.7	<0.001
Light physical activity	73.1 ± 19.7	67.3 ± 17.0	79.3 ± 20.5	<0.001
Sedentary physical activity	291.9 ± 37.5	301.7 ± 35.7	281.2 ± 36.5	<0.001
Physical activity during leisure time (mean ± SD)				
Moderate-to-vigorous physical activity	31.6 ± 15.0	30.9 ± 13.8	32.3 ± 16.1	0.145
Light physical activity	73.0 ± 22.6	75.4 ± 21.5	70.5 ± 23.5	<0.001
Sedentary physical activity	324.8 ± 69.3	330.0 ± 63.5	319.2 ± 74.7	0.011
Reached the recommendation for physical activity				
Yes	273 (30.2)	121 (24.7)	152 (36.8)	<0.001
No	630 (69.8)	369 (75.3)	261 (63.2)	
School Level Variables				
Number of schools	34 (100)	N/A	N/A	
Type of municipality				
Rural	4 (11.8)	N/A	N/A	
Urban	30 (88.2)	N/A	N/A	
Proportion of parents with high education (mean ± SD)	59.9 ± 16.9	N/A	N/A	

* According to the International Obesity Task Force age-standardised cut-offs from 2012. Sig: Significant.

**Table 2 ijerph-18-10239-t002:** Associations between the school environment and the participants’ moderate-to-vigorous physical activity (MVPA) in minutes during school time.

	All	Girls	Boys	Interactions with Sex
	β (95% CI)	β (95% CI)	β (95% CI)	*p*
From teachers				
Health policy (n school/n students)				
No (consistent) (17/300)	REF	REF	REF	**0.007**
Conflicting answers (7/166)	**2.56 (0.54, 4.41)**	0.79 (−1.53, 3.13)	**5.00 (1.11, 8.39)**	
Yes (consistent) (10/300)	**3.87 (2.37, 5.23)**	**2.55 (0.67, 4.33)**	**4.67 (2.66, 6.70)**	
Mobile ban				
No (consistent and mixed) (11/330)	REF	REF	REF	0.347
Yes (consistent) (23/581)	**2.51 (1.29, 3.94)**	**2.27 (0.62, 3.70)**	**2.26 (0.32, 4.43)**	
Activity breaks during lessons				
No (consistent) (11/307)	REF	REF	REF	0.996
Conflicting answers (8/282)	**4.71 (3.19, 6.28)**	**4.33 (2.53, 6.14)**	**4.96 (2.27, 7.26)**	
Yes (consistent) (14/305)	−0.46 (−2.01, 1.07)	0.16 (−1.88, 2.36)	−0.19 (−3.14, 2.37)	
Organized physical activity during breaks				
No (consistent) (16/469)	REF	REF	REF	0.922
Conflicting answers (9/255)	1.18 (−2.92, 1.01)	0.41 (−1.17, 2.19)	1.18 (−1.28, 3.56)	
Yes (consistent) (8/165)	−0.84 (−0.34, 2.63)	−1.27 (−3.63, 0.76)	−1.55 (−4.36, 1.24)	
From school schedules				
Break variables (n students)				
Total duration per day (911)	−0.00 (−0.10, 0.10)	**0.12 (0.01, 0.23)**	−0.06 (−0.22, 0.04)	0.451
Average duration of breaks (911)	−0.03 (−0.36, 0.29)	0.23 (−0.15, 0.63)	−0.35 (−0.78, 0.05)	0.131
Number of breaks per day (911)	−0.55 (−2.55, 1.41)	1.49 (−0.83, 4.03)	−0.69 (−4.30, 1.38)	0.626
Physical education				
Total duration per week (911)	**0.08 (0.05, 0.11)**	**0.08 (0.04, 0.12)**	**0.10 (0.05, 0.14)**	**0.030**
Average duration of physical education classes (911)	**0.08 (0.01, 0.13)**	0.05 (−0.02, 0.12)	**0.09 (0.005, 0.17)**	**0.039**
Number of physical education classes per week (911)	**1.73 (0.50, 3.04)**	**1.95 (0.55, 3.37)**	**2.21 (0.54, 4.61)**	0.947

Models are adjusted for sex, wear time and parental education and school clustering. Results in bold are significant at α < 0.05.

**Table 3 ijerph-18-10239-t003:** Associations between the school environment and the participants’ light physical activity (LIPA) in minutes during school time.

	All	Girls	Boys	Interactions with Sex
	β (95% CI)	β (95% CI)	β (95% CI)	*p*
From teachers				
Health policy (n school/n students)				
No (consistent) (17/300)	REF	REF	REF	**0.002**
Conflicting answers (7/166)	0.60 (−3.08, 4.56)	−2.26 (−6.05, 1.40)	3.78 (−2.74, 10.67)	
Yes (consistent) (10/300)	2.32 (−0.23, 5.16)	−1.87 (−4.49, 1.13)	5.43 (0.98, 9.70)	
Mobile ban				
No (consistent and mixed) (11/330)	REF	REF	REF	0.053
Yes (consistent) (23/581)	1.03 (−1.26, 3.21)	−1.12 (−3.59, 1.48)	2.21 (−2.30, 5.98)	
Activity breaks during lessons				
No (consistent) (11/307)	REF	REF	REF	**<0.001**
Conflicting answers (8/282)	0.12 (−2.77, 3.18)	**−4.57 (−8.19, −0.92)**	**5.26 (0.94, 9.46)**	
Yes (consistent) (14/305)	−0.78 (−3.57, 2.26)	**−4.04 (−7.35, −0.76)**	3.41 (−1.55, 8.63)	
Organized physical activity during breaks				
No (consistent) (16/469)	REF	REF	REF	0.055
Conflicting answers (9/255)	2.74 (−0.19, 5.31)	3.36 (−0.09, 6.31)	1.60 (−3.16, 565)	
Yes (consistent) (8/165)	**4.92 (1.65, 8.72)**	2.40 (−1.35, 7.06)	**7.68 (1.88, 13.92)**	
From school schedules				
Break variables (n students)				
Total duration per day (911)	0.06 (−0.07, 0.21)	0.01 (−0.13, 0.16)	0.15 (−0.04, 0.36)	0.114
Average duration of breaks (911)	0.34 (−0.24, 0.79)	0.29 (−0.39, 0.92)	0.58 (−0.15, 1.28)	0.379
Number of breaks per day (911)	−1.44 (−3.52, 1.20)	−2.24 (−4.45, 0.28)	0.04 (−3.82, 3.68)	0.223
Physical education				
Total duration per week (911)	**0.06 (0.004, 0.12)**	0.004 (−0.06, 0.07)	**0.10 (0.01, 0.18)**	**0.025**
Average duration of physical education classes (911)	0.06 (−0.03, 0.14)	0.005 (−0.11, 0.11)	0.08 (−0.04, 0.18)	0.203
Number of physical education classes per week (911)	0.95 (−1.28, 3.34)	0.17 (−2.89, 3.00)	1.80 (−1.39, 5.17)	0.663

Models are adjusted for sex, wear time and parental education and school clustering. Results in bold are significant at α < 0.05.

**Table 4 ijerph-18-10239-t004:** Associations between the school environment and the participants’ sedentary time in minutes during school time.

	All	Girls	Boys	Interactions with Sex
	β (95% CI)	β (95% CI)	β (95% CI)	*p*
From teachers				
Health policy (n school/n students)				
No (consistent) (17/300)	REF	REF	REF	**0.001**
Conflicting answers (7/166)	−3.25 (−8.44, 1.35)	1.34 (−4.28, 6.63)	**−8.74 (−18.02, −0.82)**	
Yes (consistent) (10/300)	**−6.41 (−10.24, −2.67)**	−0.89 (−4.44, 2.80)	**−9.95 (−15.37, −5.12)**	
Mobile ban				
No (consistent and mixed) (11/330)	REF	REF	REF	0.058
Yes (consistent) (23/581)	**−3.75 (−7.25, −0.77)**	−1.29 (−4.72, 2.02)	−4.45 (−9.32, 1.77)	
Activity breaks during lessons				
No (consistent) (11/307)	REF	REF	REF	**0.006**
Conflicting answers (8/282)	**−4.50 (−7.87, −1.17)**	0.58 (−3.66, 4.22)	**−9.62 (−16.24, −3.89)**	
Yes (consistent) (14/305)	1.54 (−2.25, 5.21)	3.84 (−0.06, 7.93)	−2.90 (−9.77, 3.63)	
Organized physical activity during breaks				
No (consistent) (16/469)	REF	REF	REF	0.153
Conflicting answers (9/255)	−3.71 (−6.94, 0.03)	−3.52 (−7.46, 0.86)	−2.55 (−8.47, 3.24)	
Yes (consistent) (8/165)	−4.42 (−9.38, 0.53)	−1.67 (−7.18, 3.67)	−6.35 (−14.56, 0.85)	
From school schedules				
Break variables (n students)				
Total duration per day (911)	−0.08 (−0.27, 0.11)	−0.13 (−0.37, 0.10)	−0.14 (−0.40, 0.16)	0.368
Average duration of breaks (911)	−0.21 (−1.00, 0.55)	−0.40 (−1.44, 0.52)	−0.23 (−1.26, 0.80)	0.921
Number of breaks per day (911)	1.13 (−2.13, 4.54)	0.53 (−2.87, 4.32)	−0.53 (−4.86, 4.77)	0.281
Physical education				
Total duration per week (911)	**−0.15 (−0.23, −0.08)**	**−0.09 (−0.17, −0.002)**	**−0.20 (−0.31, −0.11)**	**0.008**
Average duration of physical education classes (911)	**−0.14, (−0.27, −0.03)**	−0.04 (−0.22, 0.11)	**−0.19 (−0.33, −0.03)**	0.057
Number of physical education classes per week (911)	−2.68 (−5.68, 0.16)	−2.19 (−5.74, 1.23)	−3.93 (−7.93, 0.66)	0.799

Models are adjusted for sex, wear time and parental education and school clustering. Results in bold are significant at α < 0.05.

## Data Availability

The datasets are not available for download in order to protect the confidentiality of the participants. The data are held at the Swedish School of Sport and Health Sciences, Sweden.

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
