# Peer review of "Associations between the School Environment and Physical Activity Pattern during School Time in Swedish Adolescents"

_ijerph, 2021, doi:10.3390/ijerph181910239_

Round 1
Reviewer 1 Report
In this manuscript, an investigation of associations between variables pertaining to the school environment and variables pertaining to physical activity and sedentary behavior in Swedish adolescents is presented. The topic falls squarely within the domain covered by this journal. The large sample and the high quality of the writing are desirable features of the manuscript. These positive impressions notwithstanding, I have several extremely minor suggestions about the manuscript in its current form:
- The comma after “accelerometry” on l. 41 should be deleted.
- On l. 121, it should be “days were considered.”
- On l. 132, it should be “during the weekend.”
- On l. 149, it should be “questions on their school environment.”
- On l. 216, it should be “Bootstrapping estimation…” (“The” should be deleted).
- On l. 218 and 222, it should be “analyses” instead of “analyzes.”
- On l. 221, the word “The” should be deleted.
- On l. 307, should it be “systematic review” instead of “study”?
9. I appreciate inclusion of the section on limitations on l. 377-383.
Author Response
Thank you for your positive and valuable comments that we believe have improved the manuscript. As requested, we have responded to the comments.
In this manuscript, an investigation of associations between variables pertaining to the school environment and variables pertaining to physical activity and sedentary behavior in Swedish adolescents is presented. The topic falls squarely within the domain covered by this journal. The large sample and the high quality of the writing are desirable features of the manuscript. These positive impressions notwithstanding, I have several extremely minor suggestions about the manuscript in its current form:
#1 The comma after “accelerometry” on l. 41 should be deleted.
- Thank you, this has been corrected on line 41.
#2 On l. 121, it should be “days were considered.”
- This has been corrected on line 125.
#3 On l. 132, it should be “during the weekend.”
- This has been corrected on line on line 137.
#4 On l. 149, it should be “questions on their school environment.”
- This has been corrected on line 155.
#5 On l. 216, it should be “Bootstrapping estimation…” (“The” should be deleted).
- This has been corrected on line 222.
#6 On l. 218 and 222, it should be “analyses” instead of “analyzes.”
- This has been corrected on lines 219, 224, 226 and 228.
#7 On l. 221, the word “The” should be deleted.
- This has been corrected on line 224.
#8 On l. 307, should it be “systematic review” instead of “study”?
- Thank you, that is correct and this has been corrected on line 310.
#9 I appreciate inclusion of the section on limitations on l. 377-383.
- Thank you for the comment.
Reviewer 2 Report
This study aimed to explore the associations between factors in the school environment, PA levels, and sedentary time during school time. The authors used the cross-section design and recruited 1139 adolescents participating in the study. PA and sedentary time were measured with accelerometers. Factors in school environment were reported by teachers. The multi-level linear regression models were employed to determine the associations. The results showed that adolescents attending schools with health policies, mobile phone bans and more time for PE classes showed higher PA levels and less time in sedentary compared to schools without policies. This is a well designed study. The authors used appropriate instruments and procedures to collect data and conducted reasonable statistics analysis. The results are reliable and the conclusions have practical meaning for the future study. Except for these strengths, here are two minor concerns:
- In "2.3 Data collection", line 95. The authors mentioned using half a day for data collection, however, there must be one week to collect the PA and sedentary time. When and who conducted the collection of PA and sedentary time?
- Line 142. Suggest provide the scale's brand name and manufacturer.
Author Response
Thank you for your positive and valuable comments that we believe have improved the manuscript. As requested, we have responded to the comments.
This study aimed to explore the associations between factors in the school environment, PA levels, and sedentary time during school time. The authors used the cross-section design and recruited 1139 adolescents participating in the study. PA and sedentary time were measured with accelerometers. Factors in school environment were reported by teachers. The multi-level linear regression models were employed to determine the associations. The results showed that adolescents attending schools with health policies, mobile phone bans and more time for PE classes showed higher PA levels and less time in sedentary compared to schools without policies. This is a well designed study. The authors used appropriate instruments and procedures to collect data and conducted reasonable statistics analysis. The results are reliable and the conclusions have practical meaning for the future study. Except for these strengths, here are two minor concerns:
#1 In "2.3 Data collection", line 95. The authors mentioned using half a day for data collection, however, there must be one week to collect the PA and sedentary time. When and who conducted the collection of PA and sedentary time?
- We agree and this has now been clarified in lines 97-104.
#2 Line 142. Suggest provide the scale's brand name and manufacturer.
- Thank you for highlighting this. We have added that in lines 147-148.